# Changes in Adenosine Deaminase Activity and Endothelial Dysfunction after Mild Coronavirus Disease-2019

**DOI:** 10.3390/ijms241713140

**Published:** 2023-08-24

**Authors:** Agata Jedrzejewska, Ada Kawecka, Alicja Braczko, Marzena Romanowska-Kocejko, Klaudia Stawarska, Milena Deptuła, Małgorzata Zawrzykraj, Marika Franczak, Oliwia Krol, Gabriela Harasim, Iga Walczak, Michał Pikuła, Marcin Hellmann, Barbara Kutryb-Zając

**Affiliations:** 1Department of Biochemistry, Medical University of Gdansk, 80-211 Gdansk, Poland; agata.jedrzejewska@gumed.edu.pl (A.J.); ada.kawecka@gumed.edu.pl (A.K.); alicja.braczko@gumed.edu.pl (A.B.); ks99@gumed.edu.pl (K.S.); marika.franczak@gumed.edu.pl (M.F.); oliwia.krol@gumed.edu.pl (O.K.); gabriela.harasim@gumed.edu.pl (G.H.); iga.walczak@gumed.edu.pl (I.W.); 2Department of Cardiac Diagnostics, Medical University of Gdansk, 80-210 Gdansk, Poland; marzena.romanowska-kocejko@gumed.edu.pl (M.R.-K.); marcin.hellmann@gumed.edu.pl (M.H.); 3Laboratory of Tissue Engineering and Regenerative Medicine, Division of Embryology, Medical University of Gdansk, 80-211 Gdansk, Poland; milena.deptula@gumed.edu.pl (M.D.); michal.pikula@gumed.edu.pl (M.P.); 4Division of Clinical Anatomy, Department of Anatomy, Medical University of Gdansk, 80-210 Gdansk, Poland; malgorzata.zawrzykraj@gumed.edu.pl

**Keywords:** COVID-19, microvascular dysfunction, endothelium, adenosine deaminase

## Abstract

Endothelial cells are a preferential target for SARS-CoV-2 infection. Previously, we have reported that vascular adenosine deaminase 1 (ADA1) may serve as a biomarker of endothelial activation and vascular inflammation, while ADA2 plays a critical role in monocyte and macrophage function. In this study, we investigated the activities of circulating ADA isoenzymes in patients 8 weeks after mild COVID-19 and related them to the parameters of inflammation and microvascular/endothelial function. Post-COVID patients revealed microvascular dysfunction associated with the changes in circulating parameters of endothelial dysfunction and inflammatory activation. Interestingly, serum total ADA and ADA2 activities were diminished in post-COVID patients, while ADA1 remained unchanged in comparison to healthy controls without a prior diagnosis of SARS-CoV-2 infection. While serum ADA1 activity tended to positively correspond with the parameters of endothelial activation and inflammation, sICAM-1 and TNFα, serum ADA2 activity correlated with IL-10. Simultaneously, post-COVID patients had lower circulating levels of ADA1-anchoring protein, CD26, that may serve as an alternative receptor for virus binding. This suggests that after the infection CD26 is rather maintained in cell-attached form, enabling ADA1 complexing. This study points to the possible role of ADA isoenzymes in cardiovascular complications after mild COVID-19.

## 1. Introduction

Coronavirus Disease-2019 (COVID-19) is an inflammatory disease caused by infection with Severe Acute Respiratory Syndrome Coronavirus 2 (SARS-CoV-2) [1]. So far, nearly 7 billion people worldwide have been infected with the SARS-CoV-2 virus, and almost 7.0 million among them died (covid19.who.int). Several kinds of vaccines have been put into use, all with proven effectiveness and safety [2,3]. Clinical manifestations of the disease can range from asymptomatic to life-threatening acute respiratory distress syndrome [4,5]. The pathophysiology of COVID-19 primarily involves a pulmonary macrophage activation syndrome, an endothelial inflammation (endotheliitis), and a procoagulant state with thrombotic microangiopathy [6]. In many cases, especially during the pre-vaccination period the pathological process was not limited to the expansion in tissues of the respiratory tract, but took the form of a severe systemic infection with a hyperinflammatory and hypercoagulable state [7,8].

It has been found that endothelial cells are a preferential target for the SARS-CoV-2 virus resulting in widespread endotheliitis [9]. In this setting, morphological and functional changes in the endothelium are critical in COVID-19-related inflammation [10,11]. It has been found that the binding of the SARS-CoV-2 virus to endothelial angiotensin-converting enzyme-2 (ACE2) receptors activates endothelial exocytosis of P-selectin, vWF, and other proinflammatory cytokines that prompt platelet aggregation and leukocyte influx into the vessel wall, which triggers the inflammation and thrombosis that particularly affect the microcirculation [12,13]. Indeed, microvascular disturbances and microthrombosis are the leading cause of COVID-19-related acute and long-term complications [14].

Previously, we have reported that adenosine-converting ecto-enzyme, which is a cell surface adenosine deaminase (ADA), may serve as a biomarker of endothelial activation and vascular inflammation in cardiovascular pathologies [15], neurodegeneration [16], and cancer [17]. Cell-attached ADA is responsible for the irreversible deamination of extracellular adenosine that plays a critical role in the maintenance of vascular homeostasis, providing anti-inflammatory, antithrombotic, and endothelial-protective effects [18]. It has been demonstrated that lower blood adenosine concentration was related to a worse prognosis of SARS-CoV-2 infection, whereas administration of exogenous adenosine prevented inflammatory responses in the COVID-19-patient-derived leukocytes in vitro [19]. In line with that, targeting the adenosine-specific receptors was proposed as an alternative approach to attenuate lung inflammation and thrombosis as consequences of COVID-19 [20].

There are two genetically distinct ADA isoenzymes, ADA1 and ADA2, which are characterized by different catalytic properties and distribution in mammalian tissues. ADA1 catalyzes the hydrolytic deamination of the amino group of adenosine with the Km value in the micromolar range and optimal pH around 7.0, while ADA2 has 100-fold less affinity for adenosine, acting in slightly acidic pH. Although both isoenzymes exist in soluble (circulating), cell-attached, and intracellular forms, ADA1 is crucial inside the cells for 2′-deoxyadenosine deamination, preventing self-cytotoxicity [21]. Impaired levels of ADA1 in humans is a cause of severe combined immunodeficiency (SCID), manifested by severe lymphopenia, diminished function, and differentiation of T and B cells [22]. In turn, intracellular ADA plays a minor role in adenosine catabolism inside the cells, as adenosine kinase (AK) with a much higher affinity for adenosine preferentially phosphorylates adenosine to adenosine monophosphate (AMP) [23]. Nonetheless, adenosine-deaminating catalytic activities of both ADA1 and ADA2 are virtually important in circulating body fluids as well on the cell surfaces [24]. It has been well described that extracellular adenosine concentration rises in response to hypoxia or inflammation, providing an adaptive mechanism to protect injured cells [25]. However, due to the rapid catabolism by ADA1 and ADA2, this process is transient. In humans, ADA1 was identified on the surface of many cell types, including endothelium, lymphocytes, macrophages, platelets, fibroblasts, erythrocytes, epithelial, and dendritic cells [26]. This so-called ecto-ADA (eADA) is attached to the cell surface by ADA-binding proteins (ADA-BP), including CD26 protein or adenosine receptors [24,27]. Interestingly CD26, also named dipeptidyl peptidase 4 (DPP-4), was considered, besides the ACE2 receptor, as a potential receptor for binding and the entrance of SARS-CoV-2 [28]. The presence of cell-attached ADA2 is much less studied, but it has been proposed that membrane proteoglycans of immune cells may be involved in ADA2 membrane docking [29]. Importantly, soluble ADA2 is a predominant ADA isoenzyme in human plasma [30]. This is caused by the secretion of ADA2 primarily from monocytes and macrophages [31]. Insufficient levels of ADA2 may cause a number of adverse clinical conditions, namely childhood-onset stroke, systemic vasculitis, variable immunodeficiency, and hematologic defects [32].

Our recent studies revealed a highly increased activity of ADA1 on the surface of proinflammatory-stimulated endothelial cells [33,34]. In later stages of vascular inflammation, both ADA isoenzymes were found in vessel tissues and predominantly originated from the inflammatory infiltrate during atherosclerosis progression [33]. Cellular compartmentation of individual ADA isoenzymes revealed that endothelial and immune cells were a rich source of intracellular and cell-surface ADA1, while ADA2 was abundant only inside the monocytes and macrophages [34]. Interestingly, we identified JAK/STAT pathway as critical for proinflammatory upregulation of endothelial ADA1 expression [34], which underlines the potential interplay between ADA and COVID-19 pathophysiology [35]. Thus, the current study aimed to analyze the activities of circulating ADA isoenzymes in patients after mild or asymptomatic COVID-19 and to link them with the parameters of endothelial and microvascular dysfunction.

## 2. Results

### 2.1. Circulating Activities of Adenosine Deaminase Isoenzymes and CD26 Protein Level in Serum Are Deregulated in Post-COVID Patients

First, we measured the activities of total adenosine deaminase (tADA) and its isoenzymes in the serum of post-COVID patients and non-COVID-19 individuals. The study revealed a decreased total ADA activity in post-COVID patients’ serum (Figure 1A), which was associated with a decreased ADA2 activity (Figure 1C), while ADA1 activity remained unchanged (Figure 1B). Further, we measured the serum level of CD26 protein, which was reduced as well in comparison to healthy patients (Figure 1D).

### 2.2. Microvascular and Endothelial Dysfunction with Proinflammatory Phenotype in Post-COVID Patients

Microvascular function analysis (Figure 2A) revealed decreased ischemic response parameters in post-COVID patients (Figure 2B), and no significant changes in maximal hyperemic response and its index (Figure 2C). In turn, microcirculatory response to hypoxia expressed as hypoxia sensitivity parameters and reactive hyperemia response, which is particularly linked to NO-dependent endothelial function, were considerably decreased in post-COVID patients (Figure 2D).

To confirm endothelial dysfunction, the amino acids’ profile was analyzed in serum (Figure 3A–H). Metabolites such as ornithine, ADMA, and L-NMMA exhibited increased concentrations in post-COVID patients, whereas the concentrations of arginine, citrulline, and SDMA were unaltered. Thus, the ornithine/arginine ratio was subsequently enhanced, while the arginine/ADMA ratio was diminished. Several other amino acids’ concentrations were measured to verify possible differences between control and post-COVID patients’ sera. The study presented a significant increase in cystine, glutamate, glutamine, glycine, lysine, methionine, and valine (Appendix A) in post-COVID patients’ sera. The other tested amino acids remained unchanged.

Additionally, we investigated several circulating parameters of endothelial activation and inflammation (Figure 4A–E). Post-COVID patients revealed no statistical differences in serum hsCRP and IL-10. In turn, TNFα and sICAM-1 concentrations were higher in the post-COVID group. This was reflected by the increased TNFα/IL-10 ratio in post-COVID patients.

In correlation analyses, we observed that serum ADA1 activity tended to positively correspond with the parameters of endothelial activation and inflammation, such as sICAM-1, TNFα, and negatively with IL-10 (Figure 5). In contrast, serum ADA2 negatively correlated with sICAM-1 and TNFα, while a strong positive relationship was observed between ADA2 and IL-10 (Figure 6).

### 2.3. Adenosine Deaminase 1 Is Overactivated on the Surface of Endothelial Cells Stimulated with Post-COVID Patient Sera

Next, we evaluated the activity of ecto-ADA isoenzymes in human lung microvascular endothelial cells (HULEC) after treating them with patients’ sera (Figure 7A). Interestingly, total cell-surface ADA activity was higher after post-COVID serum treatment as a result of an increase in cell-surface ADA1 activity. Ecto-ADA2 isoenzyme activity was virtually unchanged. After treatment, the concentration of cell protein as well as intracellular adenosine triphosphate (ATP) and nicotinamide adenine dinucleotide (NAD) were at similar levels in both groups (Figure 7B,C). Then, to analyze leukocyte-endothelial interactions, we investigated the adhesion of neutrophil-like HL-60 cells to endothelial monolayer after the exposition on patients’ sera (Figure 7D,E). HULEC pretreated with post-COVID serum revealed the increased adhesion of HL-60 cells in comparison to these treated with control serum.

## 3. Discussion

The current paper points to the possible role of ADA isoenzymes in cardiovascular complications after mild COVID-19. In patients, we demonstrated a long-term microvascular and endothelial dysfunction together with the decreased concentration of serum ADA1-complexing protein CD26 and ADA2 isoenzyme activity. Although minor changes in circulating ADA1 activity were revealed, we speculate that after the infection, ADA1 and CD26 are rather maintained in cell-attached form, deactivating local adenosine-related pathways. On the other hand, we observed significant negative correlations between serum ADA2 activity and proinflammatory phenotype reflected by increased levels of circulating adhesion molecules, TNFα concentration, and TNFα/IL10 ratio, which is assumed as an independent predictor for coronary artery disease. It was demonstrated previously that ADA2 is crucial for M2 monocyte/macrophage polarization and, hence, its decreased activity may be related to the disturbed proportion of circulating monocyte subtypes [36,37]. Furthermore, we observed a trend to lower platelet count in post-COVID patients, which may also be connected with higher level of serum TNFα. Not long after the outbreak of COVID-19, multiple case reports and case series underlining the importance of monitoring post-COVID patients have recognized secondary immune thrombocytopenia (ITP) onset or relapse as one of the viral infection complication [38,39]. Among the reasons underlying the observed decrease in platelet count below 100 × 10^9^/L were listed damage of hematopoietic tissues and megakaryocytes caused by interaction of SARS-CoV-2 with ACE2, CD13, and CD66 present on their surface [40]. The following inflammatory response, causing cytokine storm and lung endothelium damage, contributes to decreased platelet production [41,42]. In our study we observed higher levels of serum TNFα in post-COVID patients, which has been positively associated with impaired megakaryocyte maturation [43].

It has been demonstrated that ADA exerts an important physiological role in the regulation of adenosine effects on immunological, neurological, and vascular processes [16,33,44]. As it is critical for the proliferation and differentiation of lymphocytes, particularly T cells, increased serum ADA activity can be seen in diseases associated with cellular immune response, such as liver disease, typhoid, infectious mononucleosis, sarcoidosis, brucellosis, pneumonia, rheumatoid arthritis, tuberculosis, and certain malignancies, especially those of hematopoietic origin [45,46,47,48,49]. Thus, the analysis of total ADA activity and particularly its ADA1 isoenzyme in serum could be useful as noninvasive diagnostic tool in clinical conditions associated with T cell stimulation. Interestingly, in our study we did not observe differences in serum ADA1 activities in post-COVID-19. Despite that, the activity of circulating ADA1 seemed to be closely related to the presence of elevated concentrations of proinflammatory cytokines or the parameters of endothelial activation in patients.

It has been well demonstrated that coordinated immune response is essential to control SARS-CoV-2 infection [50]. The induction of innate immunity (e.g., monocytes/macrophages, neutrophils, and dendritic cells) results in the production of various proinflammatory cytokines and chemokines, which helps to clear the virus [51]. However, overproduction of these mediators can lead to cytokine storm and lung injury [52]. In turn, the presentation of antigen by antigen-presenting cells induces humoral or cellular immunity responses [53]. Specific CD4^+^ cells became activated and differentiated into Th cells. Subsequently, they are engaged in virus clearance by the production of macrophage-activating factors, such as TNFα by Th1 cells, and via antibody production by B cells under Th2 cell activation. In turn, CD8^+^ cells differentiate into cytotoxic T cells that contribute to virus clearance by the lysis of infected cells [54]. In severe COVID-19, activated phenotype in CD8^+^ and CD4^+^ T cells have been found as compared to mild disease manifestation and noninfected normal controls [55,56]. In line with that, it was found that serum ADA activity was two times higher in patients with severe COVID-19 in comparison to moderate course of the disease [57]. Of note, patients with moderate SARS-CoV-2 infection also had higher total serum ADA activity than healthy individuals, but the samples were collected during the time of the symptoms and/or positive PCR test results [57]. Moreover, it has been found that ADA can be increased in various body fluids during infectious and inflammatory states [58]. It is no different in COVID-19 where ADA activity was higher in saliva of symptomatic COVID-19 and 10 days after positive SARS-CoV-2 PCR than in healthy controls [59]. In another study, the nasopharyngeal specimen of SARS-CoV-2 infected patients revealed higher ADA expression level in nasopharyngeal swabs when compared with controls and, interestingly, when compared to the swabs obtained from patients with severe COVID-19 [60]. This suggests an essential role of ADA in respiratory track local manifestation of the disease and the therapeutic potential of its inhibition. However, suppressing ADA to augment protective adenosine was proposed to be favorable rather in improving long-term chronic clinical outcomes in COVID-19 patients than throughout the early phase of the infection [61]. Deoxycoformycin (dCF), a potent and tightly binding ADA inhibitor was proposed to balance the signals through adenosine receptors as beneficial in late stage acute respiratory distress syndrome [61,62]. In turn, our previous studies revealed protective effects of dCF on endothelial activation and inflammation upregulating adenosine-dependent receptor effects [34].

This study demonstrated that even in the presence of microvascular and endothelial dysfunction followed by mild COVID-19 as well as during persistently elevated levels of proinflammatory cytokines and parameters of endothelial activation, such as TNFα and sICAM-1, serum ADA1 is not increased. This is in contrast to our previous results where ADA1 activity was closely related to endothelial activation and vascular inflammation [33]. However, it should be noted that in our later study, it was rather membrane-anchored ADA1 that marked endothelial cell activation, while changes in soluble ADA were less pronounced in experimental models of vasculitis or in human specimen [33]. This ecto-ADA is bound to the cell surface by ADA-complexing proteins that include CD26 protein and A1 or A2a adenosine receptors [24,26]. Interestingly, circulating CD26 protein concentration was diminished in our patients 8 weeks after mild COVID-19 in comparison to controls that was in line with other studies [63]. Moreover, even lower levels of CD26 in serum were demonstrated in patients with severe COVID-19 [64]. This suggests that during the SARS-CoV-2 infection and in a few weeks after CD26 is rather maintained in cell-attached form, enabling ADA1 binding. Our in vitro studies revealed that lung microvascular endothelial cells exposed to post-COVID patients’ sera upregulated cell-surface ADA1 activity. Augmented endothelial eADA expression may explain the lack of its increase in serum and lower concentration of circulating CD26 protein. In addition, it may have significant consequences resulting in the depletion of extracellular adenosine levels. The silencing of adenosine signaling may largely contribute to the vascular complications observed in our patients that are associated with microvascular dysfunction and endothelial inflammation [65].

Since CD26 was considered as an alternative receptor for SARS-CoV-2 binding [28], the anchoring of ADA1 to CD26 may disturb cell entrance of the virus. It has been shown that ADA competitively attaches to CD26, thereby preventing the binding of the Middle East Respiratory Syndrome Coronavirus (MERS-CoV) spike protein domain to CD26 [66]. Thus, we assume that ADA1 can be recognized as a natural antagonist that blocks viral attack and ameliorates a severe course of SARS-CoV-2 infection, but at the same time, it may be a cause of long-term vascular complications after mild/moderate COVID-19 due to adenosine signaling modulation and its role as an adhesion molecule. Indeed, we revealed that endothelial cells exposed in vitro on post-COVID patients’ sera switched into the proinflammatory phenotype, increasing their potential to neutrophil adhesion. Therefore, in addition to the analysis of CD26 concentration as a predictor of severe outcomes of COVID-19, we suggest the assessment of microvascular function and in vitro functional tests using patients’ sera to predict late vascular complication and proinflammatory endothelial phenotype. It would also be particularly valuable to investigate it in already vaccinated patients. On the other hand, the administration of the competitors for the docking process of SARS-CoV-2 to CD26 may counteract the long-term negative effects derived from ecto-ADA enzymatic and nonenzymatic properties together with the prevention of viral entry. It was evidenced that treatment with sitagliptin, the inhibitor of CD26 enzymatic activity as dipeptidyl peptidase-4, reduced mortality in patients hospitalized for COVID-19 [67]. It is plausible that the protective effects of dipeptidyl peptidase-4 inhibitors are mediated by the interference with the SARS-CoV-2-CD26 interaction [67,68].

In conclusion, this study describes endothelial activation after mild COVID-19 and points to possible involvement of ADA isoenzymes in cardiovascular complications after recovery. Our results suggest that after the infection, CD26 is rather maintained in cell-attached form, enabling ADA1 complexing. Observed lower activity of ADA2 isoenzyme in post-COVID serum, which closely positively correlated with anti-inflammatory IL-10 concentration, could be connected to M2 polarization. As we aimed in this study to characterize the relation of circulating ADA isoenzyme pattern with endothelial and microvascular dysfunction in patients after mild COVID-19, further research on the mechanical explanations of our findings are highly requested. The limitation of our outcomes poses a relatively small cohort of post-COVID patients and healthy control groups. The control was evaluated at the beginning of the pandemic, when anti-COVID vaccines were still unavailable and we had confidence none of the patients from the control group had gone through asymptomatic COVID-19 nor had been vaccinated against it prior to being tested. We propose that further studies in vitro on ADA1 competition with SARS-CoV-2 for the binding to endothelial CD26 protein could provide more information on the mechanical circumstances of these interactions. The correlation between ADA isoenzyme pattern in the serum of post-COVID patients and the proportion of circulating monocytes should be further examined in order to assess whether ADA2 could be used as an indicator of deregulated proinflammatory cytokine production in post-COVID-19 individuals.

## 4. Materials and Methods

### 4.1. Human Participants

All patients provided written informed consent in line with the Declaration of Helsinki, and the approval for the study was granted by the Independent Bioethics Committee for Scientific Research at the Medical University of Gdansk, Poland under the license number NKBBN/55/2021. Whole blood was collected from patients after asymptomatic or mild COVID-19, 8 weeks after returning a positive PCR test for SARS-CoV-2 (post-COVID) and from healthy individuals, without prior diagnosis of SARS-CoV-2 infection (controls). The samples were collected during the pre-vaccination period, in February–March 2021. Thus, none of the subjects enrolled in the study received any vaccine dose. Whole blood was centrifuged at 1200× *g* for 15 min at room temperature to obtain serum. Another blood sample was taken into a tube containing the anticoagulant heparin and centrifuged at 1200× *g* for 15 min at room temperature. The obtained serum was secured after centrifugation and immediately frozen at −80 °C for later analyses.

### 4.2. Measurement of Microvascular Function

Microvascular function was assessed in post-COVID-19 patients characterized above (Table 1) and non-COVID-19 individuals (Appendix A) using flow-mediated skin fluorescence (FMSF). FMSF is a noninvasive optical technique used to evaluate microcirculation and metabolic regulation based on the measurements of NADH fluorescence intensity in epidermis [69]. Quantification of FMSF was performed using AngioExpert created by Angionica Ltd. (Lodz, Poland) as described earlier [70].

Several parameters were measured during NADH fluorescence registration: the ischemic response (IR max), IR index, and HR index, all described by Pajkowski et al. [71]. The direct measurements of oscillations during the reperfusion stage allow us to assess the hypoxia sensitivity (HS), which covers the intensity of flow motion connected to myogenic oscillations. The reactive hyperemia response (RHR) parameter was the sum of the IR max and HR max parameters and reflected vascular endothelial function related to nitric oxide production in blood vessels, due to occlusion-induced hyperemia.

### 4.3. Quantification of Amino Acids in Serum

Serum amino acid concentrations were determined using LC/MS as previously described [72]. Briefly, an aliquot of plasma (50 µL) was enriched with internal standards and extracted using 100 µL of acetonitrile for 15 min on ice. Subsequently, the samples were centrifuged at 4 °C, 20,800× *g* for 10 min. The supernatants were collected and freeze-dried. The obtained sediments were dissolved in 100 µL of distilled water and analyzed by using ion-pair high-performance liquid chromatography with mass detection in positive mode electrospray ionization. To identify individual amino acids, their molecular weight, chromatographic retention time, and fragmentation pattern were used as was described earlier.

### 4.4. Quantification of hsCRP, sICAM-1, CD26 Protein, and Cytokins’ Concentration in Serum and Peripheral Blood Morphology

The concentration of serum high-sensitive C-reactive protein (hsCRP) and peripheral blood morphology were measured using standard methods. CD26 and sICAM-1 concentration was determined in serum using ELISA kits according to manufacturer’s instructions (Merck, Darmstadt, Germany). TNFα and IL-10 concentration was measured in serum samples by Luminex Multiplex platform using dedicated kits (Merck, Darmstadt, Germany) according to manufacturer’s instructions.

### 4.5. Measurement of Adenosine Deaminase Activity in Serum

In order to assess the activities of soluble total ADA (tADA) and ADA2, 49 μL of serum was brought to 37 °C prior to the experiment and adenosine to the final concentration of 20 μM was added, with or without ADA1 inhibitor, erythro-9-(2-hydroxy-3-nonyl) adenine (5 μM EHNA). Following 30 min incubation, the reaction was terminated via deproteinization with 1.3 M HClO_4_ at 1:1 ratio. After 15 min incubation on ice, the samples were centrifuged (20,800× *g*, 4 °C, 15 min). The supernatant was brought to pH 6.5–7.0 with 3M K_3_PO_4_ and analyzed using the UHPLC system as described earlier [16].

### 4.6. Cell Culture

Human microvascular lung endothelial cells (HULEC-5a) were obtained from ATCC (Manassas, VA, USA) and cultured in MCDB131 medium supplemented with 10% fetal bovine serum (FBS), 10 ng/mL Epidermal Growth Factor (EGF), 1 μg/mL hydrocortisone, 10 mM L-glutamine, and 1% penicillin/streptomycin. Human neutrophils (HL-60) were obtained from ECACC (Salisbury, UK) and cultured in RPMI 1640 medium supplemented with 2 mM L-glutamine, 10% fetal bovine serum and 1% penicillin/streptomycin. All cultured cells were maintained at 37 °C, 5% CO_2_. For the treatments, HULEC cells were seeded in 24-well plates (5 × 10^4^ cells/well) or 96-well plates (0.8 × 10^4^ cells/well) and after reaching 80% confluence the cells were washed 3 times with PBS to discard any remaining traces of FBS form the initial culture media, and then incubated for 48 h with MCDB131 medium free of FBS supplemented with 20% serum obtained from post-COVID patients (N = 3) or controls (N = 3).

### 4.7. Measurement of Adenosine Deaminase Activity on Endothelial Cell Surface

HULEC monolayer was treated as described above in 24-well plates and rinsed with PBS. Then, 1 mL of Hanks Balanced Salt Solution (HBSS) was added to each well. To measure cell-surface total adenosine deaminating activity, 50 μM adenosine was added and samples were collected after 0, 5, 15, and 30 min of incubation at 37 °C and analyzed using UHPLC as described previously [16]. To measure ADA2 activity, the above assay was conducted in the presence of 10 μM erythro-9-(2-hydroxy-3-nonyl)adenine (EHNA). ADA1 activity was calculated by subtracting the activity of ADA2 from a total adenosine deaminating activity.

### 4.8. Immune Cell Adhesion Assay

To study HL-60 adhesion to endothelial cells, HULEC cells were seeded in 24-well cell imaging plates (Merck Millipore, Darmstadt, Germany) and treated as described above. After washing with PBS, carboxyfluorescin succinimidyl ester (CFSE)-labeled HL-60 cells (4.5 × 10^4^) were added at 300 µL of FBS-free medium. Cells were co-incubated for 30 min at 37 °C, 5% CO_2_ and then nonadhered HL-60 cells were removed and plates were rinsed twice with 1 mL FBS-free medium. Then, HL-60 cell adhesion to endothelial monolayers was measured as an area of residual green fluorescence staining using an Axio Observer 7 fluorescence microscope and ZEN software v.3.3 blue edition (Carl Zeiss Inc., Dresden, Germany).

### 4.9. Statistical Analysis

Statistical evaluation of the obtained data was carried out using InStat software (GraphPad Prism 9.0, San Diego, CA, USA). Normality was determined using the following tests: Kolmonotov–Smirnov test, Shapiro–Wilk test, or D’Agostino and Pearson Omnibus test. The mean values between groups were compared using a one-way Anova and, subsequently, a Holm–Sidak or Dunn’s post hoc test with an unpaired Student’s *t*-test or Mann–Whitney test. The correlations were analyzed using Pearson’s correlation coefficient. The precise value of n was given for each experiment. Statistical significance was assigned at *p* < 0.05.

## Figures and Tables

**Figure 1 ijms-24-13140-f001:**
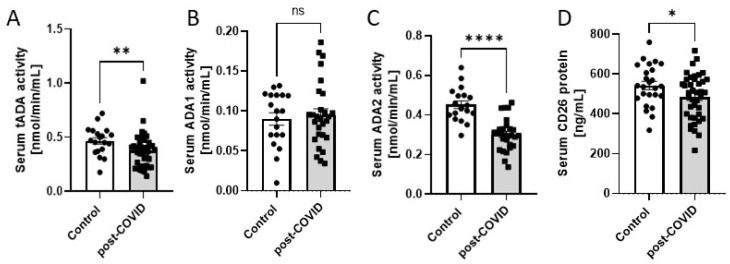
(**A**) The activities of total adenosine deaminase (tADA); (**B**,**C**) adenosine deaminase isoenzymes (ADA1 and ADA2); (**D**) CD26 protein concentration in serum of post-COVID patients (n = 40) and controls (n = 25). Results are shown as mean ± SEM, ns—not significant, * *p* < 0.05, ** *p* < 0.01, and **** *p* < 0.0001 according to Student’s *t*-test.

**Figure 2 ijms-24-13140-f002:**
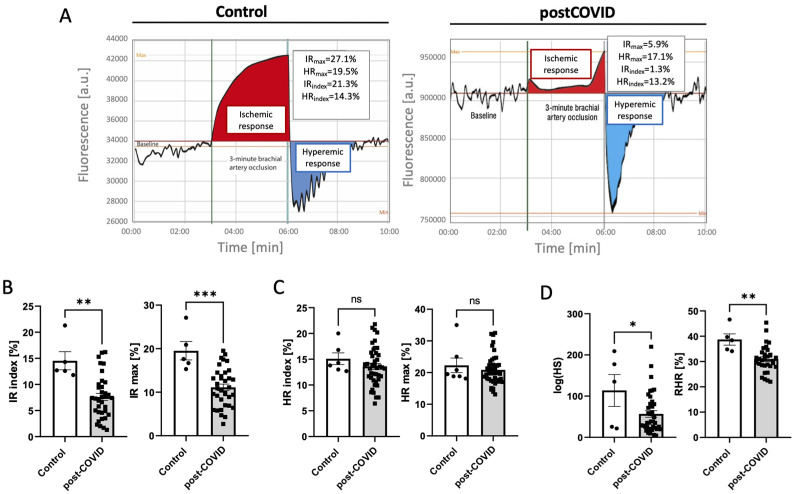
(**A**) Representative images of nicotinamide adenine dinucleotide (NADH) fluorescence trace in response to blockage and release of blood flow in the brachial artery analyzed by flow-mediated skin fluorescence (FMSF) in control and post-COVID patients. Ischemic response (IR) parameters (**B**), hyperemic response (HR) parameters (**C**), hypoxia sensitivity (log(HS) and reactive hyperemia response (RHR) (**D**) assessed according to FMSF in post0-COVID patients (n = 40) and controls (n = 5). Results are shown as mean ± SEM, ns—not significant, * *p* < 0.05, ** *p* < 0.01, and *** *p* < 0.001 related to Student’s *t*-test or the Mann–Whitney test as appropriate.

**Figure 3 ijms-24-13140-f003:**
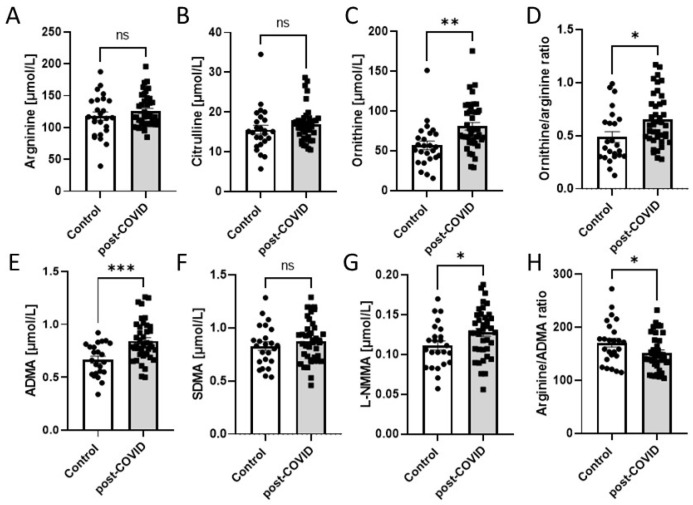
Circulating L-arginine (**A**), L-citrulline (**B**), L-ornithine (**C**), L-ornitine/L-arginine ratio (**D**), asymmetric dimethyl-L-arginine (ADMA) (**E**), symmetric dimethyl-L-arginine (SDMA) (**F**), N-monomethyl-L-arginine (L-NMMA) (**G**), L-arginine/ADMA ratio (**H**) in serum of post-COVID patients (n = 40) and controls (n = 25). Results are shown as mean ± SEM, ns—not significant, * *p* < 0.05, ** *p* < 0.01, *** *p* < 0.001 according to Student’s *t*-test.

**Figure 4 ijms-24-13140-f004:**
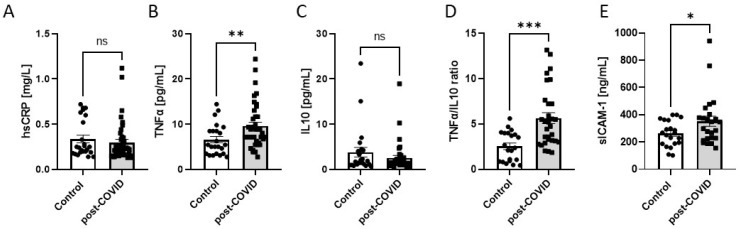
Circulating CRP (**A**), TNFα (**B**), IL10 (**C**), TNFα/IL-10 ratio (**D**), and soluble ICAM-1 (sICAM1) (**E**) in serum of post-COVID patients (n = 40) and controls (n = 25). Results are shown as mean ± SEM, ns—not significant,* *p* < 0.05, ** *p* < 0.01, *** *p* < 0.001 according to Student’s *t*-test.

**Figure 5 ijms-24-13140-f005:**
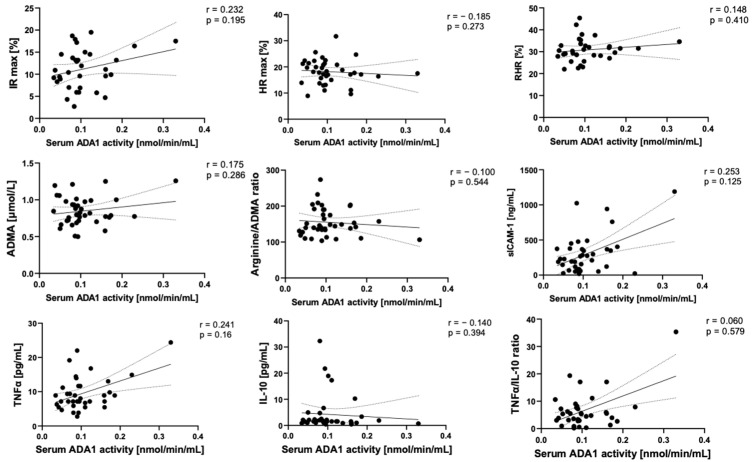
Correlation plots of serum ADA1 activity with maximal ischemic response (IR max), maximal hyperemic response (HR max), and reactive hyperemic response (RHR) assessed using the FMSF technique, serum concentration of asymmetric dimethyl-L-arginine (ADMA), L-arginine/ADMA ratio, soluble ICAM-1 (sICAM-1), TNFα, IL10, and TNFα/IL-10 ratio with corresponding Spearman r coefficient and *p* value. Solid line—regression line, dotted line—error bars.

**Figure 6 ijms-24-13140-f006:**
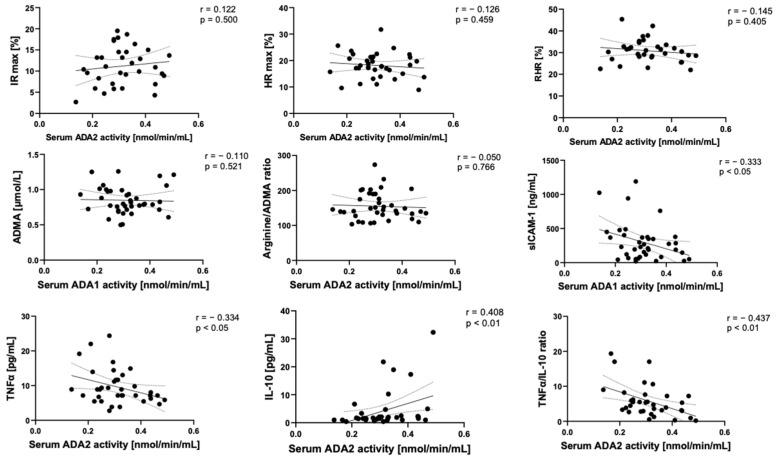
Correlation plots of serum ADA2 activity with maximal ischemic response (IR max), maximal hyperemic response (HR max), and reactive hyperemic response (RHR) assessed using the FMSF technique, serum concentration of asymmetric dimethyl-L-arginine (ADMA), L-arginine/ADMA ratio, soluble ICAM-1 (sICAM-1), TNFα, IL10, and TNFα/IL-10 ratio with corresponding Spearman r coefficient and *p* value. Solid line—regression line, dotted line—error bars.

**Figure 7 ijms-24-13140-f007:**
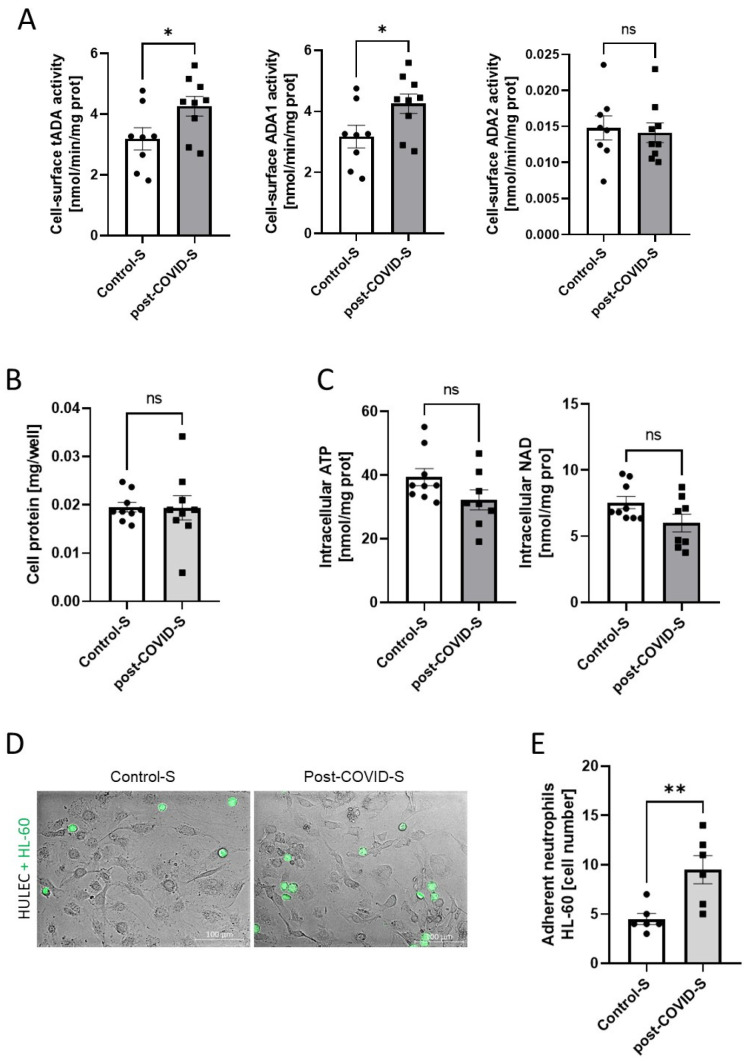
(**A**) Cell-surface total ADA (tADA), ADA1, and ADA2 activities; (**B**) Cell protein concentration; (**C**) Intracellular adenosine triphosphate (ATP) and nicotinamide adenine dinucleotide (NAD) concentration in human lung microvascular endothelial cells (HULEC) treated for 48 h with control (Control-S) and post-COVID patients’ (postCOVID-S) sera; n = 3 independent experiments with 3 different patients’ sera, n = 3 biological repetitions for each patient. (**D**) Representative images (**D**) and Quantitative analysis (**E**) of the adhesion of HL-60 cells (green) to HULEC pre-treated for 48 h with control (Control-S) and post-COVID patients’ (post-COVID-S) sera. Results are shown as mean ± SEM, ns—not significant,* *p* < 0.05, ** *p* < 0.01 according Student’s *t*-test.

**Table 1 ijms-24-13140-t001:** Post-COVID patient characteristic. Results are shown as mean ± SD, BMI—body mass index, SBP—systolic blood pressure, DBP—diastolic blood pressure, HR—heart rate, ACE—angiotensin converting enzyme, na.—not applicable. Hb—hemoglobin, HCT—hematocrit, RBC—red blood cells, WBC—white blood cells, PLT—platelets.

Parameter	Controls (n = 25)	postCOVID (n = 40)
Age (years)	40.0 ± 12.5	40.5 ± 8.7
Female	n = 18 (72%)	n = 27 (67.5%)
BMI (kg/m^2^)	23.4 ± 3.8	25.5 ± 4.9
SBP/DBP (mmHg)	123/78 ± 9.0/5.2	125/85 ± 15.6/9.2
HR (beats/min)	76 ± 9.0	77 ± 11.4
Hospitalization	na.	n = 1 (2.5%)
Symptoms		
Fatigue	na.	n = 26 (65%)
Tachycardia	na.	n = 14 (35%)
Chest pains	na.	n = 14 (35%)
Breathing difficulties	na.	n = 5 (12.5%)
Pharmacotherapy		
Beta-adrenolitics	na.	n = 13 (32.5%)
ACE-inhibitors	na.	n = 5 (12.5%)
Blood morphology		
Hb [g/dL]	13.5 ± 1.5	14.2 ± 1.3
HCT [%]	40.6 ± 3.2	41.2 ± 3.4
RBC [10^12^/L]	4.62 ± 0.5	4.70 ± 0.4
WBC [10^9^/L]	6.10 ± 2.4	6.49 ± 2.2
Neutrophils [10^9^/L]	3.77 ± 1.5	3.72 ± 1.6
% Neutrophils	58.1 ± 12	56.0 ± 9.1
Lymphocytes [10^9^/L]	2.46 ± 0.9	2.07 ± 0.6
% Lymphocytes	37.9 ± 10	33.1 ± 8.5
Monocytes [10^9^/L]	0.48 ± 0.2	0.50 ± 0.2
% Monocytes	7.40 ± 2.5	7.98 ± 2.1
Eosinophils [10^9^/L]	0.12 ± 0.05	0.16 ± 0.12
% Eosynophils	1.85 ± 0.8	2.40 ± 1.2
Basophils [10^9^/L]	0.04 ± 0.02	0.04 ± 0.02
% Basophils	0.62 ± 0.2	0.57 ± 0.2
PLT [10^9^/L]	276 ± 90	230 ± 51

## Data Availability

The data presented in this study are available on request from corresponding author.

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
