# Peer review of "Changes in Adenosine Deaminase Activity and Endothelial Dysfunction after Mild Coronavirus Disease-2019"

_ijms, 2023, doi:10.3390/ijms241713140_

Round 1

Reviewer 1 Report

It is known that endothelial cells are a primarily target for the SARS-CoV-2 virus resulting in systemic endothelial inflammation. Accordingly, morphological and functional changes in the endothelium are detected to be critical in COVID-19-related inflammation, where microvascular disturbances and microthrombosis are recognized as the leading cause of COVID-19-related short-term and long-term complications.

The authors argue that in later stages of vascular inflammation, both ADA isoenzymes were found in vessel tissues, mainly originated from the inflammatory infiltrate during atherosclerosis progression. So, the submitted study aimed to analyze the activities of circulating ADA isoenzymes in patients after mild or asymptomatic COVID-19, and to link them with the parameters of endothelial and microvascular dysfunction.

The authors presented clear and valuable data in regard to endothelial dysfunction and SARS-CoV-2 infection, especially knowing that the vast majority of papers were linked to fully developed, or life threatening disease, but not the mild cases. In this context there is a clear added value from the clinical point of view. The methodology was properly chosen and adequately used, with obtained results having clear significance that were, moreover, suitably discussed in regard to the postulated scientific purpose.

The authors point to the possible role of ADA isoenzymes in cardiovascular complications after mild COVID-19, where a long-term microvascular and endothelial dysfunction together with the decreased concentration of serum ADA1-complexing protein CD26 and ADA2 isoenzyme activity was demonstrated.

Finally, through the adequate identification of the limits, the authors suggest that the correlation between ADA isoenzyme pattern in the serum of postCOVID patients and the proportion of circulating monocytes should be further examined in order to assess whether ADA2 could be used as an indicator of deregulated pro-inflammatory cytokine production in postCOVID-19 individuals. I fully support this idea, as well.

/

Author Response

Thank you for reviewing and appreciation of our manuscript.

Reviewer 2 Report

This is a well written paper on Covid19 infection and its effects on microcirculation and biomarkers of endothelial function and inflammation.

As is, the paper reads well and the results are well supported by solid research. As is I have no further comments.

Author Response

Thank you for reviewing our work and allowing it to be published.

Reviewer 3 Report

The manuscript by Jedrzejewska et al described that among 40 patients 8 weeks after mild COVID-19 infection, serum adenosine deaminase 2 (ADA2) activity was decreased, along with increase in markers of inflammation and endothelial dysfunction. The authors proposed that ADA could play a role in increasing risk of cardiovascular complications in post-acute COVID-19 patients.

Overall, the paper is focusing on an important topic, especially as reports of post-COVID cardiovascular complications continue to emerge. The authors have strong track records in the relevant studies on ADA and its role in inflammatory vascular diseases. The study is well designed, investigating multiple protein markers and metabolites in drawn blood and microcirculation in human subjects using PMSF. The following points are for the considerations of the authors:

1. The main limitation of the current study is probably the relatively small cohort size of 40 post-COVID patients (25 controls for blood markers, 5 controls for PMSF). Although this is fine as a pilot/exploratory study. It would be great for the readers if the authors can write a paragraph in the discussion to evaluate this and any other additional limitations for their study.

2. Related to the above, with the small cohort, the bar charts in Fig. 1 to 4 would be a lot more informative if presented as box plot or dot plot that can show data distribution.

3. Although ‘8 weeks after asymptomatic/mild COVID-19’ is stated in the graphical abstract, the information on distribution of the number of days post infection among the subjects is not available. Please add this information into the manuscript. Please also add other details if available eg. if any patients were hospitalized, any repeat infections, how many were infected before vaccinated, how many after being vaccinated, physical/clinical well-being post-COVID (ie. if patients experiencing any long COVID or cardiovascular symptoms or the inflammatory state is sub-clinical?). I understand many of these information may not be available depending on the study design but any additional information to better characterize the cohort would be extremely useful.

4. In Table 1, platelet counts for the post-COVID group is noticeably lower – did the authors test for statistical significance? Can the authors comment on this?

Author Response

Thank you for reviewing our manuscript. We addressed your comments and suggestions.

Ad. 1 As suggested, the limitations have been added to the discussion (l.332-340).

Ad. 2 We changed the charts on dot plots with bars (Fig. 1-4).

Ad. 3 Information about the time that has passed since the positive SARS-CoV-2 PCR test has been added to the methodology (l.352-356). Similarly, as data became available, the characterization of the group of patients was expanded (Table 1). In addition, we strengthened the work with in vitro experiments showing that the stimulation of lung microvascular endothelial cells with serum from post-COVID patients increases the activity of adenosine deaminase 1 on their surface as well as stimulates the pro-inflammatory potential of endothelial cells (Figure 7).

Ad. 4 In our analysis, platelets counts between groups were not statistically different. After unpaired t-test p value was 0.633. The results in the table are presented as the mean + - SEM. Nevertheless, the trend of lower platelets is evident and has been discussed in the paper in the discussion section (l.228-238)